# In Vitro versus in Mice: Efficacy and Safety of Decoquinate and Quinoline-*O*-Carbamate Derivatives against Experimental Infection with *Neospora caninum* Tachyzoites

**DOI:** 10.3390/pathogens12030447

**Published:** 2023-03-13

**Authors:** Jessica Ramseier, Dennis Imhof, Kai Pascal Alexander Hänggeli, Nicoleta Anghel, Ghalia Boubaker, Richard M. Beteck, Luis-Miguel Ortega-Mora, Richard K. Haynes, Andrew Hemphill

**Affiliations:** 1Institute of Parasitology, University of Bern, Länggass-Strasse 122, 30132 Bern, Switzerland; 2Graduate School for Cellular and Biomedical Sciences, University of Bern, Mittelstrasse 43, 3013 Bern, Switzerland; 3Centre of Excellence for Pharmaceutical Sciences, Faculty of Health Sciences, North-West University, Potchefstroom 2520, South Africa; 4SALUVET, Animal Health Department, Faculty of Veterinary Sciences, Complutense University of Madrid, Ciudad Universitaria s/n, 28040 Madrid, Spain

**Keywords:** decoquinate, *O*-quinoline carbamates, *Neospora caninum*, in vitro, mice, safety, efficacy

## Abstract

The effects of decoquinate (DCQ) and three *O*-quinoline-carbamate-derivatives were investigated using human foreskin fibroblasts (HFF) infected with *Neospora caninum* tachyzoites. These compounds exhibited half-maximal proliferation inhibition (IC_50_s) from 1.7 (RMB060) to 60 nM (RMB055). Conversely, when applied at 5 (DCQ, RMB054) or 10µM (RMB055, RMB060), HFF viability was not affected. Treatments of infected cell cultures at 0.5µM altered the ultrastructure of the parasite mitochondrion and cytoplasm within 24 h, most pronounced for RMB060, and DCQ, RMB054 and RMB060 did not impair the viability of splenocytes from naïve mice. Long-term treatments of *N. caninum*-infected HFF monolayers with 0.5µM of each compound showed that only exposure to RMB060 over a period of six consecutive days had a parasiticidal effect, while the other compounds were not able to kill all tachyzoites in vitro. Thus, DCQ and RMB060 were comparatively assessed in the pregnant neosporosis mouse model. The oral application of these compounds suspended in corn oil at 10 mg/kg/day for 5 d resulted in a decreased fertility rate and litter size in the DCQ group, whereas reproductive parameters were not altered by RMB060 treatment. However, both compounds failed to protect mice from cerebral infection and did not prevent vertical transmission/pup mortality. Thus, despite the promising in vitro efficacy and safety characteristics of DCQ and DCQ-derivatives, proof of concept for activity against neosporosis could not be demonstrated in the murine model.

## 1. Introduction

Infection by the apicomplexan parasite *Neospora caninum* is one of the major causes of abortion and stillbirth in dairy and beef cattle, and thus has a great economic impact worldwide [1]. Canids are the definitive hosts wherein sexual development takes place, with many warm-blooded animals serving as intermediate hosts. Three infective stages are found in the life cycle of the parasite: (i) tachyzoites, which represent the rapidly multiplying stage and cause acute disease; (ii) slowly proliferating bradyzoites that form tissue cysts and persist for years without clinical signs; and (iii) oocysts, containing sporozoites encapsulated in sporocysts. The formation of oocysts takes place in the intestinal tissue of canids, which shed them into the environment via their feces. Infection can take place by the ingestion of tissues infected with tachyzoites or tissue cysts, or by the ingestion of food or water contaminated with sporulated oocysts. Most importantly, the temporal immunomodulation that occurs during pregnancy allows for the vertical transmission of tachyzoites from the dam to its fetus, either during primo-infection in a pregnant animal (exogenous transplacental transmission), or through recrudescence of the quiescent bradyzoites that re-differentiate into tachyzoites (endogenous transplacental transmission) [1]. *N. caninum* tachyzoites can infect virtually any cell type, a feature they share with the closely related *Toxoplasma gondii*. However, in contrast to *T. gondii*, no cases of human infection by *N. caninum* have been reported so far [1]. Besides causing disease in pregnant cattle, neosporosis can lead to neonatal complications and neuromuscular diseases in dogs [2,3].

To date, safe and efficient drugs against neosporosis in farm animals are lacking [4]. For the treatment of canine neosporosis, the lincosamid clindamycin, often combined with synergistically acting sulfonamides and pyrimethamine, represent the primary treatment option. However, whilst these treatments inhibit tachyzoites, they do not affect the bradyzoite stage, and the effects of are most pronounced early after infection [3]. Treatments have to be taken over long periods, and for dogs with neurological signs, the prognosis is poor. In addition, clindamycin treatment in horses, cattle, sheep and goats may cause severe diarrhea that can result in death. Furthermore, no vaccine against *N. caninum* is on the market, either for canids or farm animals [4]. The repurposing of drugs originally developed for other indications has been applied for identifying novel chemotherapeutic options.

The quinolone decoquinate (DCQ, ethyl 6-decyloxy-7-ethoxy-4-oxo-1*H*-quinoline-3-carboxylate, Figure 1A) is marketed worldwide as a compound to treat gastrointestinal coccidiosis in poultry, cattle and small ruminants. For the treatment, it is normally formulated at 0.5–1 mg/kg body weight and added to the feed [5]. DCQ acts as a mitochondrial cytochrome *bc*_1_ inhibitor, interfering in the electron transfer from ubiquinone to cytochrome *c* and inhibiting oxidative phosphorylation and thus ATP production [6]. While DCQ is generally regarded to be safe in ruminants, its physicochemical properties such as high lipophilicity and exceedingly low aqueous solubility (0.06 μg/mL) limits the absorption and bio-availability of the compound [6]. DCQ was previously shown to be active against *N. caninum* tachyzoites in vitro at concentrations greater than 0.01 µg/mL [7]. In pregnant sheep treated with DCQ from 10 days prior to oral challenge with *T. gondii* oocysts at 90 days of gestation until lambing, DCQ reduced placental damage inflicted by infection, and increased the gestation period and the number and weight of live lambs in comparison to non-treated ewes challenged with *T gondii* oocysts [8]. Thus, we investigated whether DCQ treatment would also be active in mice experimentally infected with *N. caninum* tachyzoites.

Due to the poor drug-like properties of DCQ, derivatives with enhanced water solubility were generated by adding highly polar groups to the initial compound [9,10]. To treat systemic infections more successfully, in our case, DCQ was derivatized to the quinoline-*O*-carbamate derivatives RMB054, RMB055 and RMB060 (Figure 1B–D). The addition of polar groups was thought to increase absorption, with the appendages metabolized in the liver following uptake and resulting in increased DCQ levels after that. These derivatives showed low cytotoxicity against human foreskin and fetal lung fibroblasts in vitro, as well as half maximal effective concentrations (IC_50_ values) against *N. caninum* tachyzoites in the nanomolar range [11]. In addition, these DCQ derivatives were also active against tachyzoites of the closely related apicomplexan *Toxoplasma gondii* in vitro. In pregnant CD1 outbred mice infected with *T. gondii* oocysts, treatment with RMB060 was safe in pregnant and non-pregnant animals, and reduced the parasite burden in non-pregnant mice but not dams. However, RMB060 treatment did not prevent the vertical transmission of *T. gondii* to pups [12].

In this study, we demonstrate the in vitro efficacy of DCQ, RMB054, RMB055 and RMB060 against *N. caninum* tachyzoites that are grown in human foreskin fibroblast monolayers as host cells, and also study the effects of these drugs on the viability of non-infected fibroblasts. Tachyzoites represent the disease-causing stage of *N. caninum*. We examined ultrastructural changes induced upon drug exposure by transmission electron microscopy (TEM). The potential impact of DCQ, RMB054 and RMB060 against proliferating immune cells were assessed in murine splenocyte cultures. DCQ and the most efficacious derivative RMB060 were further evaluated in vivo using pregnant BALB/c mice experimentally infected with the virulent *N. caninum* strain NcSp-7. However, DCQ and RMB060 failed to exert protective effects in vivo.

## 2. Materials and Methods

### 2.1. Cell Culture Equipment and Media, Biochemicals and Compounds

Cell culture devices were purchased from Sarstedt (Sevelen, Switzerland), the biochemicals from acquired from Sigma (St. Louis, MO, USA), and the culture media came from Gibco-BRL (Zürich, Switzerland). The DCQ was kindly provided by Prof. Gilles Gasser, Chimie ParisTech—PSL, University of Paris. RMB054, RMB055, and RMB060 were synthesized and purified as described previously [11]. All compounds were received as powder. For in vitro studies, stock solutions of 1 and 10 mM were prepared in dimethyl-sulfoxide (DMSO) and stored at −20 °C, and for in vivo experiments, compounds were resuspended in corn oil at the concentrations indicated below.

### 2.2. Host Cells and Parasites

Human foreskin fibroblasts (HFF; PCS-201-010TM) were cultured as described by Ramseier et al. [12]. Tachyzoites of *N. caninum* β-gal (Nc-β-gal) constitutively expressing β-galactosidase were kindly provided by Prof. David L. Sibley of Washington University, St. Louis, MO, USA. All parasites were maintained as previously described by Winzer et al. [13].

### 2.3. Cytotoxicity and Anti-N. caninum Efficacy Assessments In Vitro

HFF cytotoxicity and in vitro efficacy measurements against Nc-β-gal tachyzoites grown in HFF (IC_50_ determinations) were performed as described previously [11,14,15]. Briefly, 10^3^ HFF seeded into 96-well-plates were grown to confluence and infected with 10^3^ Nc- β-gal tachyzoites per well in the presence of a compound or DMSO only as a control. After 3 days of culture at 37 °C/5% CO_2_, the medium was removed, and after one wash with phosphate-buffered saline (PBS), cells were overlaid with 0.1 mL PBS containing Triton X-100 (0.05%) and chlorophenyl red-β-D-galactopyranoside (Roche, Rotkreuz, Switzerland). Absorption was continuously read at 570 nm in a spectrophotometer [12]. The IC_50_ values were calculated by logit-log-transformation of the relative growth and subsequent regression analysis using the Microsoft Excel software package (Microsoft, Seattle, WA, USA).

### 2.4. Transmission Electron Microscopy

HFF monolayers were grown in T25 culture flasks and were infected with 10^7^ *N. caninum* Spain-7 (NcSp-7) tachyzoites during 3 h at 37 °C and 5% CO_2_. Subsequently, cultures were exposed to continuous treatment with 0.5 μM of each of DCQ, RMB054 or RMB060, and 1 µM RMB055 for 6 h, 9 h, and 48 h. Control cultures were treated with DMSO only. The specimens were fixed and processed for TEM as described earlier [14,16,17]. Samples were observed on a Philips CM 12 or a FEI Tecnai Spirit BioTwin TEM, both operating at 80 kV.

### 2.5. Long Term Treatment of N. caninum Infected HFF

For longer durations of treatment, HFF grown in T25 culture flasks were infected with 1 × 10^6^ NcSp-7 tachyzoites, and treatment with each of 0.5 μM of DCQ, RMB054, or RMB060 was initiated after 4.5 h of incubation. Controls were conducted by the addition of DMSO (0.5%) to infected cultures. Prolonged treatments for up to 24 consecutive days with the addition of fresh medium containing the respective drug every 3–4 days were carried out. Cultures were inspected by light microscopy on a daily basis. On days 3, 6, 9, 13, 16, 20 and 24, the compound-containing medium was removed, and parasites were maintained without drug pressure until lysis of the monolayer due to tachyzoite proliferation was evident.

### 2.6. Assessment of Susceptibility of Murine Splenocytes to DCQ, RMB054 and RMB060

Murine splenocytes were isolated from female BALB/c mice and assessed for drug susceptibility as described [18]. A Trypan Blue dye exclusion test was performed to determine the viability of isolated cells, and only preparations with >99% viable cells were used further. The spleen cell preparation was suspended in RPMI 1640 medium including 10% FCS, 0.05 mM 2-mercaptoethanol, 2 mM L-glutamine, and 100 U of penicillin plus 50 mg of streptomycin per mL, and cells were distributed in polystyrene 96 well flat bottom sterile plastic plates (Greiner Bio-One, Kremsmünster, Austria; HuberLab, Aesch, Switzerland) at 2 × 10^5^ cells/100 μL/well. For proliferation/viability assays, splenocytes were either left unstimulated or were stimulated with Concanavalin A (ConA, 5 μg/mL), lipopolysaccharide (LPS, 10 μg/mL), ConA plus DCQ, RMB054 or RMB060 (0.1 and 0.5 µM) or LPS plus the three drugs at the same concentrations. Experiments were carried out in quadruplicates at 37 °C and 5% CO_2_ for 72 h. The proliferation of splenocytes was measured using a BrdU cell proliferation kit (QIA58, Merck Millipore, Burlington, MA, USA) according to the instructions provided by the manufacturer, and absorbance was measured at 450/540 nm in an EnSpire multilabel reader (Perkin Elmer, Waltham, MA, USA). For viability assessments, resazurin (0.1 mg/mL) was added, and the fluorescence intensity at 530/590 nm at 0, 1, 2, 3, 4, or 5 h was measured. Differences were calculated by subtracting time point 0 values from each time point. The data are presented as the mean of emission +/− standard deviation for the indicated numbers. Data comparisons between groups were conducted using a student’s *t*-test.

### 2.7. Ethics Statement

Animal experiments were approved by the Animal Welfare Committee of the Canton of Bern under the license BE117/20. The animals were handled in strict accordance with practices to minimize suffering. BALB/c mice were purchased from Charles River (Sulzberg, Germany) at 6 weeks of age. They were maintained with food and water ad libitum in a common room under controlled temperature and a 14 h dark/10 h light cycle. For adaptation, the mice were housed in the facility for two weeks prior to the experiment.

### 2.8. Assessment of the Efficacy of DCQ and RMB060 in BALB/c Mice Infected with NcSp-7 Tachyzoites

For this experiment, 42 female and 21 male BALB/c mice were used, all of them 8 weeks of age. Oestrus-synchronization of females for 3 d by the Whitten effect was carried out, and mice were mated by housing two females and one male for 72 h in one cage. Males were removed after mating and females were randomly assigned to four experimental groups: DCQ, DCQ + infection (*n* = 12); RMB060, RMB060 + infection (*n* = 12); C+, corn oil + infection (*n* = 12); and C−, only corn oil (*n* = 6). Three days prior to infection, NcSp-7 tachyzoites were transferred from HFF to BALB/c fibroblasts and maintained at 37 °C and 5% CO_2_ [19]. At day 7 post-mating, the parasites were collected from culture flasks, counted, and infection was done by subcutaneous injection with 10^5^ tachyzoites in 100 µL PBS per mouse. The C− group received only BALB/c dermal fibroblasts in PBS. Two days post-infection, treatment with DCQ or RMB060 suspended in corn oil at 10 mg/kg/day for 5 days or only with corn oil was initiated. Pregnant mice were distributed into single cages at day 18 post-mating. The non-pregnant mice were housed together (three to four animals per cage). Birth took place between day 20–22, and clinical status, litter size, neonatal and postnatal mortality were recorded daily. Four weeks after birth, all mice were euthanized in a chamber by isoflurane/CO_2_ inhalation, followed by blood, brain and eye sample collection. Total IgG was measured by ELISA [20].

### 2.9. Real-Time PCR-Based Determination of Cerebral Parasite Loads

For DNA purification, the NucleoSpin DNA RapidLyze Kit (Macherey-Nagel, Oensingen, Switzerland) was used according to the manufacturer’s protocol, and DNA concentrations were quantified by using the QuantiFluor double-stranded DNA (dsDNA) system (Promega, Madison, WI, USA). The detection of *N. caninum* to quantify the cerebral parasite load in non-pregnant mice, dams and surviving pups was performed with a TaqMan probe-based qPCR in a CFX96 qPCR instrument (BioRad Laboratories AG, Cressier, Switzerland). The qPCR is targeted to the repetitive genomic sequence of Nc5 of *N. caninum* [21]. The reaction mixture (10 µL per reaction) consisted of 5 µL 2x Mastermix (SensiFAST^TM^ Probe NO-ROX Kit; Bioline Meridian Life Science, Memphis, TN, USA), 500 nM forward primer Np21plus (5′-CCCAGTGCGTCCAATCCTGTAAC-3′), the reverse primer Np6plus (5′-CTCGCCAGTCAACCTACGTCTTCT-3′) [21], 100 nM of detection probe Nc5-1 (5′-*FAM*-CACGTATCCCACCTCTCACCGCTACCA-*BHQ-1*-3′) [22], and 5 ng of the DNA sample. Additionally, 1 unit of heat-labile Uracil DNA Glycosylase (UDG) and 300 nM dUTP (supplementary to dTTP included in the 2x Mastermix) (both from Bioline Meridian Lifesciences) were added to the reaction to remove eventual carry-over contamination from previous reactions as previously described [23]. For UDG-mediated decontaminations, an initial 10 min incubation at 40 °C followed by a 5 min denaturation step at 95 °C was included in the temperature profile. During 50 cycles of 10 s at 95 °C and 30 s at 60 °C, DNA amplification was achieved. Light emission by the fluorophore was thereby measured after each cycle. Brain samples from non-pregnant mice and dams were measured in triplicates, and pup brains as only single values. The parasite load was calculated by including external standards of DNA samples from 10,000, 1000, 100, and 10 *N. caninum* tachyzoites per run. For the analysis of the PCR results, the CFX manager software version 1.6 was used.

### 2.10. Statistical Analysis

Statistical analyses and the creation of graphs were performed using GraphPad Prism version 5.0 (GraphPad Software, La Jolla, CA, USA). Comparisons of cerebral parasite burdens between groups were conducted with the non-parametric Kruskal-Wallis test, followed by the Mann-Whitney-U test. By plotting survival events at each time point in Kaplan-Meier graphs, pup mortality over time was depicted, and survival curves were compared by the Log-rank (Mantel-Cox) test.

## 3. Results

### 3.1. In Vitro Efficacy and Safety Assessment of DCQ and Quinoline-O-Carbamate Derivatives

The in vitro efficacies of DCQ, RMB054, RMB055, and RMB060 (Figure 1A–D) were determined. The results with regard to IC_50_ values and HFF cytotoxicity are shown in Table 1, and respective dose response curves are found in Figure 1E–H. When the drugs were added concomitantly to infection, IC_50_ values were 2.4 nM for DCQ, 11.7 nM for RMB054, 60 nM for RMB055, and 1.7 nM for RMB060. However, when the compounds were added to already intracellular parasites, IC_50_ values remained in a similar range for RMB054 and RMB060, but for DCQ the value was seven-fold higher, and for RMB055 it was 2.5-times higher. The compounds did not impair the viability of uninfected HFF up to concentrations of 5 µM (DCQ/RMB054) or 10 µM (RMB055/RMB060), and no changes in the monolayer morphology were noted by light microscopy assessment (Appendix A).

DCQ, RMB054 and RMB060 were also assessed for their potential effects on immune cells. No impact on the viability or the proliferative capacities of T or B cells could be detected (Appendix A). RMB055 was excluded from this analysis due to its higher IC_50_.

### 3.2. Structural Alterations Induced by In Vitro Drug Treatments of N. caninum Infected HFF

Alterations in the parasite ultrastructure induced by these drugs were studied by TEM. NcSp-7 tachyzoites grown in the absence of any compound (Figure 2A–C) exhibited typical features of apicomplexan parasites. They were situated in the host cell within a parasitophorous vacuole, and separated from the host cell cytoplasm by a parasitophorous vacuole membrane. The apical complex consisting of conoid, rhoptries and micronemes as well as dense granules and the nucleus were clearly visible. *N. caninum* tachyzoites have a single mitochondrion exhibiting a tubule-like structure, with an electron-dense matrix and numerous cristae. Only distance areas of the mitochondrion are visible on a single section by TEM, depending on the section plane.

Treatment with 500 nM DCQ for 6 h and 9 h caused only small alterations in the mitochondria, with the matrix slightly less electron dense and partially dissolved cristae (Figure 2D,E). After 48 h of DCQ treatment, the mitochondrial matrix appeared electron-lucent, and the cristae were no longer visible. However, the mitochondrial membrane remained structurally intact, and membranous and particulate material accumulated in the interior (Figure 2F).

In contrast, treatment with 500 nM RMB060 for 6 h and 9 h had no influence on the mitochondria. The mitochondrial matrix still appeared to be quite electron-dense, and the cristae were visible (Figure 2G,H). However, after 48 h of RMB060 treatment, the mitochondrion completely lost its normal structure, the matrix disappeared, and the cristae were largely dissolved (Figure 2I). Additionally, vacuoles in the parasite cytoplasm were formed, which were either empty or filled with particulate and often electron dense material of an unknown nature. For both compounds, only one to two parasites were visible per vacuole, meaning that proliferation was slowed down. In cultures treated with RMB054 or RMB055, similar effects occurred, comparable to those obtained for DCQ (Figure 3).

### 3.3. Treatments of Extended Drug Exposure to Reveal Parasitostatic versus Parasiticidal Effects

The long-term treatment over periods of up to 24 consecutive days was carried out and followed by light microscopy for each of DCQ, RMB054 and RMB060, all applied at 0.5 µM. HFF were not affected by these treatments (Appendix A). DCQ and RMB054 only exerted a parasitostatic effect. Thus, parasites would recover and started to lyse the host cell layer at day 10–14 after drug removal, even after a treatment of 24 consecutive days (see Appendix A). In contrast, RMB060 exhibited parasiticidal activity, as no plaque formation was visible after a 6 day-treatment, even after 56 d of culture without compound (see Appendix A).

### 3.4. Efficacy of DCQ and RMB060 in BALB/c Mice Experimentally Infected with N. caninum Tachyzoites

Since long-term treatments showed that RMB054 did not act parasiticidal in vitro, this derivative was excluded from further assessments in vivo. Thus, the efficacy of RMB060 was assessed comparatively with DCQ in pregnant and non-pregnant *N. caninum* infected BALB/c mice. The results are shown in Figure 4 and Table 2.

Two dams and one non-pregnant mouse from the DCQ-treated group, three dams from the RMB060-treated group, and two dams from the C+ group showed neurological signs, including a circling motion, head tilting and twitches. One dam of the DCQ group had to be euthanized 10 days after she gave birth due to strong clinical signs of neosporosis; the remaining pups were given to another mother. Furthermore, the fertility rate and litter size of the DCQ group were decreased compared to the other groups, whereas reproductive parameters were not altered by RMB060 treatment. Overall, neonatal mortality in both treatment groups was slightly reduced compared to the placebo-(corn oil) treated group C+ (14%, 6 out of 43 pups), with 3.4% (1 out of 29 pups) in the DCQ group and 9.1% (4 out of 44 pups) in the RMB060 group. However, postnatal mortality was 100% in both treatment groups, similar to the control group C+. In C+, all pups died within 17 days p.p., while all pups succumbed to infection on day 19 p.p. in the DCQ and the RMB060 group (Figure 4A). In the adults, including non-pregnant mice and dams, brain infection was detected in 9 out of 12 mice in the DCQ-treated group, and in 11 out of 12 mice in the RMB060-treated and the C+ group (Table 2).

Non-pregnant mice did not exhibit any differences in the parasite load between treatment groups and C+ (Figure 4B). In dams, the mean values of *N. caninum* tachyzoites/µg DNA from the DCQ and the C+ group were similar, whereas RMB060 showed an even higher value. Furthermore, the DCQ group displayed a higher range of values, with the highest (5005 tachyzoites/µg DNA) in the dam that had to be euthanized earlier (Figure 4C).

IgG titers were determined from all *N. caninum* infected mice by ELISA. In all infected animals, increased IgG levels were recorded, with IgG titers all being in the same range (Appendix A). Additionally, Neospora DNA was quantified in eye samples to check for possible ocular transmission. All mice were tested as PCR-negative, meaning a transmission of *N. caninum* into the eyes can be most likely excluded (data not shown). Thus, despite promising IC_50_ values and low cytotoxicity seen in vitro, neither DCQ nor RMB060 treatment protected adult mice or pups from *N. caninum* infection.

## 4. Discussion

The quinolone DCQ is widely used for the treatment of coccidiosis caused by different *Eimeria* species, causing infections of the gastrointestinal tract in chicken, cattle and small ruminants. Other quinolones are in clinical use as antibiotics and as anti-malarials [24]. DCQ is also active in vitro against a panel of other apicomplexan parasites including invasive stages of *Sarcocystis neurona*, and *T. gondii* and *Besnoitia besnoiti* tachyzoites [7,11,12,25,26]. The closely related endochin-like quinolones (ELQs) exhibited in vitro activity against *B. besnoiti* tachyzoites [27], different *Babesia* species [28], and against *T. gondii* tachyzoites not only in vitro, but also in vivo [29,30,31]. ELQs were also highly efficacious in experimental mouse models of *N. caninum* infection [32], with improved efficacy when applied in combination with a bumped kinase inhibitor targeting calcium dependent protein kinase 1 (BKI) [33].

The mitochondrial cytochrome *bc*_1_ complex transfers electrons from coenzyme Q to cytochrome *c* [34]. As this transfer is blocked by DCQ, the relocation of electrons to other biomolecules can generate free radicals which are toxic to the parasite [35,36,37]. In general, DCQ exhibits low aqueous solubility, poor absorption, and limited bioavailability. The DCQ *O*-quinoline carbamate derivatives RMB054, RMB055 and RMB060 exhibited high activity in vitro against multi-drug-resistant strains of *P. falciparum* against *T. gondii* and *N. caninum*, and displayed no or very minor cytotoxicity in human fibroblast cultures [11]. We here applied an in vitro culture and a standardized pregnant neosporosis BALB/c mouse model to investigate the effects of DCQ and the three quinoline-*O*-carbamate derivatives RMB054, RMB055 and RMB060 against *N. caninum.*

The IC_50_s of DCQ and its derivatives against *N. caninum* tachyzoites were in the low nanomolar range (1.7–60 nM) when the drug was added concomitantly to host cell infection, similar to previously reported values [11]. However, the IC_50_ value of DCQ was approximatively seven times higher, and thus the drug was seven times less effective, when the compound was added to already infected cell cultures. In the case of RMB055, this difference was only two-fold, and for the other two compounds, no changes were noted. This indicated that DCQ and RMB055 did not affect extracellular parasites, but only impaired the proliferation of intracellular tachyzoites, while RMB054 and RMB060 exerted effects on both intracellular and extracellular parasites. Similar results had been obtained previously for *T. gondii* tachyzoites treated with these compounds [12].

The treatment of *N. caninum* infected cell cultures with 500 nM of the compounds (which corresponded to the concentration inducing complete growth inhibition in dose response assays), caused distinct ultrastructural changes within the parasite cytoplasm, depending on the drug, after 6–24 h post-treatment. These changes included alterations, especially in the matrix and the cristae, which were most dramatically altered in the case of treatment by DCQ and RMB060. Less pronounced changes were noted with the two other drugs. Similar findings were reported earlier upon the DCQ and buparvaquone treatment of *B. besnoiti* tachyzoites [25,38] and in ELQ-treated *N. caninum* tachyzoites [32,39]. Interestingly, in all these instances, parasites recovered from the initial drug effects and started to resume proliferation in vitro. The distortion of the mitochondrial cristae could constitute an escape mechanism that allows the parasite to circumvent the detrimental effects of the drug on the cytochrome *bc*_1_ complex. The shut-down of oxidative phosphorylation would then result in parasites acquiring energy through glycolysis, an aspect that deserves further investigation.

Our results using long-term drug exposure confirmed that the parasites find a way to deal with treatment with DCQ, RMB054 and RMB055, as no parasiticidal activity was observed. The molecular basis of this adaptation is not clear. On one hand, the effects seen in this study could be based on epigenetic effects, that is, adaptation without genetic changes and simply based on altered protein expression. This may be similar to what has been observed for multinucleated complexes that are formed upon the exposure of *N. caninum* tachyzoites with BKIs, inhibitors of calcium dependent protein kinase 1 [14,16,40]. This could include the differential expression of drug targets or genes involved in detrimental effects on the parasite, such as proteins involved in ROS formation or ROS scavengers [18,41], components of the detoxification machinery including ABC transporters, heat shock proteins, or glutathione transferases [42,43,44]. On the other hand, mutations within the target gene, in this case the cytochrome *b* gene, could also be responsible for the induction of resistance, similar to the case of *Theileria* treated with buparvaquone [45]. The only compound displaying parasiticidal activity when applied at 0.5 µM was the *O*-quinoline carbamate RMB060. This contrasted with the results of the long-term treatment of *T. gondii* with this and the other compounds, which had no parasiticidal effects [12].

Based on the promising in vitro safety and efficacy results obtained with RMB060, we comparatively assessed the in vivo efficacy of RMB060 in comparison with DCQ in the pregnant neosporosis mouse model. However, treatments with DCQ or RMB060 under the conditions used herein were not able to prevent parasite dissemination and vertical transmission, and did not reduce pup mortality. Overall, the DCQ treated group displayed impaired fertility compared to all other groups, which also corresponds to earlier pregnancy interference assays reported earlier [12]. Pharmacokinetic analyses in C57BL/6 mice had shown that the maximum concentrations of DCQ after oral administration of RMB060 were 0.11 ± 0.01 μM, and thus were approximately 50 times higher than the IC50 value obtained in vitro; the DCQ elimination half-life was 4.66 ± 1.16 h, and the DCQ clearance was 21.50 ± 3.38 h, respectively. Thus, RMB060 appears to act as a classical prodrug [10].

While adjustments to the treatment protocol, such as the timing of the beginning of treatment, extension of treatment duration or alterations in the dosage and formulations, could potentially improve efficacy, it is evident that the encouraging results obtained in vitro could not be translated to the murine model. In pregnant CD1 outbred mice infected with *T. gondii* oocysts, treatment with RMB060 had also not reduced vertical transmission of *T. gondii* to pups, but RMB060 treatment had exhibited partial efficacy, as it lowered the parasite burden in non-pregnant mice [12], in contrast to our study employing BALB/C mice. The reason for this discrepancy between the two models is not apparent. It is unlikely that inbred and outbred mice exhibit anatomical and physiological dissimilarities that could serve as an explanation, but immunological parameters related to infection could play an important role. It is known that pregnant BALB/C mice develop an inherent Th2-bias upon infection, while for an efficient immune response capable of keeping *N. caninum* infection in check, a mixed Th1/Th2 immune response would be required [19]. Another reason could be that different infection routes were applied in the two studies, such as the oral administration of *Toxoplasma* oocysts versus subcutaneous injection of a much higher number of *N. caninum* tachyzoites. In addition, the bioavailability, metabolic stability, systemic exposure and other pharmacokinetic properties of a compound will also influence its efficacy.

While the results of this in vivo study with the BALB/c mouse model are not encouraging, it is also important to keep in mind that murine and ruminant models are intrinsically different. Pharmacokinetic studies on milking cows that received 0.5 mg/kg DCQ indicated that the compound was well absorbed and had reached maximum plasma concentrations of 2 μM (as opposed to 0.11 µM in mice), which is several-fold higher than the exposure required to limit *N. caninum* proliferation [5]. The treatment of pregnant sheep with 2 mg/kg DCQ from 10 days prior to oral challenge with *T. gondii* oocysts at 90 days of gestation until lambing caused a delay in the onset of the febrile response to infection, reduced fever, and delayed antibody production. This reduced the placental damage caused by the parasite, increased the mean gestation period, and led to higher numbers and weights of live lambs, in comparison to the infected but non-treated control group [8]. It is possible that starting the treatment regimen prior to challenge could also have increased the efficacy against *N. caninum* infection in our experiments. To date, DCQ is marketed for the prophylaxis of *Toxoplasma*-induced abortion in pregnant ewes, with a recommended dose of 2 mg/kg/day during the last 14 weeks of pregnancy. Only one study reports on the use of DCQ in *N. caninum* infected heifers. Either chronically infected or experimentally infected heifers (*n* = 77) were treated orally with DCQ (2 mg/kg/day) from 1.5 to 8 months of pregnancy. This led to a reduction in abortion numbers and infections detected in calves at birth. However, these findings were obtained from a conference abstract only [46], and a respective publication of these findings is missing.

## 5. Conclusions

Despite being highly efficacious in vitro (IC_50_ values in the low nanomolar range and low host cell toxicity), and with parasiticidal activity noted for RMB060, both DCQ and RMB060 failed to limit parasite dissemination and fetal infection in the pregnant neosporosis mouse model. While the DCQ treatment group showed impaired fertility in our model, most likely related to drug treatment, this was not the case for RMB060. A potential way to improve in vivo efficacy could be the formulation of DCQ and/or derivatives using other vehicles, such as solid dispersions, nano/microparticles, polymeric micelles, nanosuspensions, lipid-based nanocarriers, and other formulations which have been developed for other compounds to increase solubility and improve oral bioavailability.

## Figures and Tables

**Figure 1 pathogens-12-00447-f001:**
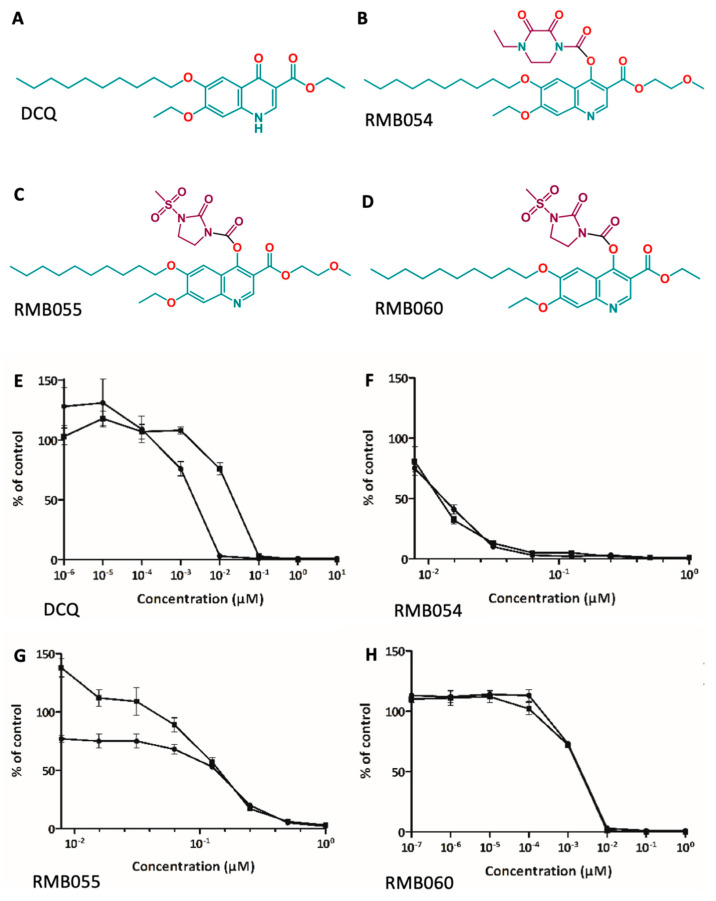
Structures and dose-response curves of decoquinate (DCQ) and its quinoline-*O*-carbamate derivatives. (**A**,**E**) DCQ, C_24_H_35_NO_5_, MW = 417.546; (**B**,**F**) RMB054, C_32_H_45_N_3_O_9_, MW = 615.724; (**C**,**G**) RMB055, C_30_H_43_N_3_O_10_S, MW = 637.745; (**D**,**H**) RMB060, C_29_H_41_N_3_O_9_S, MW = 607.719. Proliferation is indicated in relation to the control (DMSO only). The corresponding IC_50_ values are listed in Table 1.

**Figure 2 pathogens-12-00447-f002:**
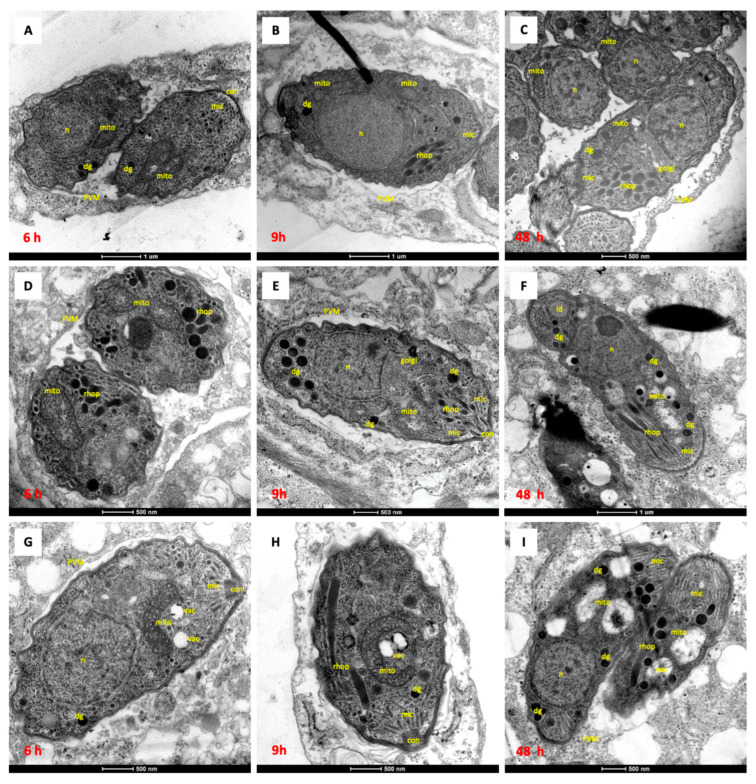
TEM of *N. caninum* Spain-7 tachyzoites cultured in HFF monolayers in vitro. HFF were infected with *N. caninum* tachyzoites and drug treatment started 3 h later. (**A**–**C**) show NcSp7 treated with DMSO as control for 6 h, 9 h and 48 h, respectively. NcSp7 treated with 0.5 µM DCQ for 6 h are shown in (**D**), 9 h in (**E**) and 48 h in (**F**). NcSp7 treated with 0.5 µM RMB060 for 6 h are shown in (**G**), 9 h in (**H**) and 48 h in (**I**). dg = dense granules; mic = micronemes; rhop = rhoptries; mito = mitochondrion; n = nucleus; PVM = parasitophorous vacuole membrane; ld = lipid droplet; vac = cytoplasmic vacuole; con = conoid.

**Figure 3 pathogens-12-00447-f003:**
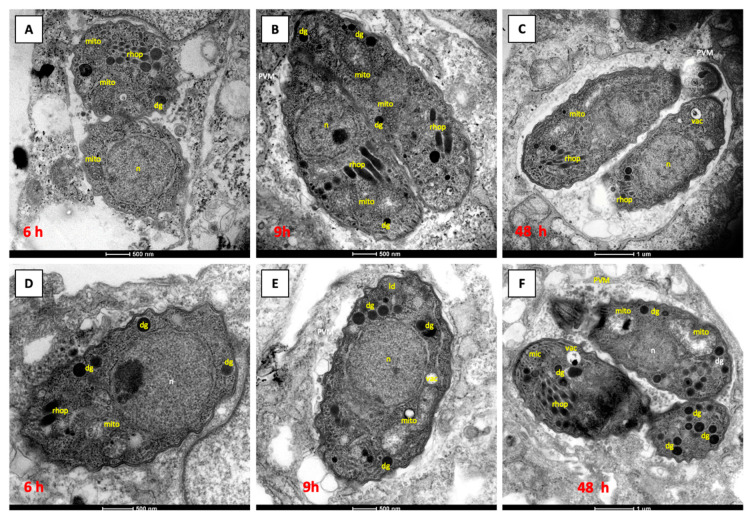
TEM of *N. caninum* Spain-7 tachyzoites cultured in HFF host cells. HFF monolayers were infected with *N. caninum* tachyzoites, followed by drug treatment 3 h later. NcSp7 treated with 0.5 µM RMB054 for 6 h are shown in (**A**), 9 h in (**B**) and 48 h in (**C**). NcSp7 treated with 1 µM RMB055 for 6 h are shown in (**D**), 9 h in (**E**), and 48 h in (**F**). dg = dense granules; mic = micronemes; rhop = rhoptries; ld = lipid droplet; mito = mitochondrion; n = nucleus; PVM = parasitophorous vacuole membrane; vac = cytoplasmic vacuole.

**Figure 4 pathogens-12-00447-f004:**
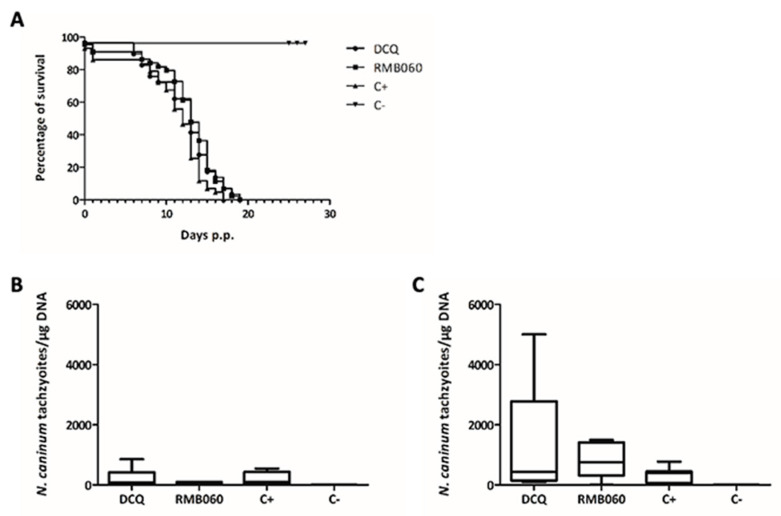
DCQ and RMB060 treatment in the pregnant *N. caninum* infected mice. BALB/c mice were infected with 10^5^ NcSp7 tachyzoites and treated with 10 mg/kg DCQ or RMB060, while C+ was infected and treated with corn oil alone. C− was not infected and treated only with corn oil. Survival curves of pups are shown in (**A**); the cerebral parasite burden of non-pregnant mice in (**B**), and dams in (**C**). (**A**) Survival rates were plotted at each time point in Kaplan-Meier graphs. No statistically significant differences between the DCQ, RMB060 and the positive control (C+) group could be observed. Four weeks after birth, all surviving mice were euthanized. The brains were collected and the cerebral parasite burden was determined by real-time PCR. Results for non-pregnant mice are shown in B, and dams in C. The parasite burdens are depicted as box plots. In both treatment groups, no statistically significant differences in the cerebral parasite burdens were observed compared to the C+ group (either in non-pregnant mice or in dams).

**Table 1 pathogens-12-00447-t001:** In vitro activities (IC_50_s) against *N. caninum* and cytotoxicity against HFF.

Compounds	IC_50_ *N. caninum* (nM) [LS; LI] ^a^ Compound Added Prior to Infection	IC_50_ *N. caninum* (nM) [LS; LI] ^a^ Compound Added after Infection	IC_50_ HFF (µM)
DCQ	2.4 [18.5; 0.3]	16.6 [52.4; 5.3]	>5 ^b^
RMB054	11.7 [18.7; 7.4]	12.7 [18.8; 8.6]	>5 ^b^
RMB055	60 [77.1; 46.7]	146.4 [161.7; 132.6]	>10 ^b^
RMB060	1.7 [6.6; 0.5]	1.2 [3.6; 0.4]	>10 ^b^

^a^ Values at 95% confidence interval (CI); LS (limit superior) and LI (limit inferior) are the upper and lower limits of the CI, respectively. ^b^ Values cannot be calculated as there is no decline in viability over the concentration range.

**Table 2 pathogens-12-00447-t002:** Litter size, parasite burden, neonatal and postnatal mortality rates of *N. caninum* infected mice treated with DCQ and RMB060.

Treatment	Challenge	*N. caninum* Seropositive	*N. caninum* Brain Positive Non-Pregnant	*N. caninum* Brain Positive Dams	Number of Pups/Dams	Neonatal Mortality	Postnatal Mortality	*N. caninum* Brain Positive Pups ^a^
10 mg/kg DCQ	10^5^ NcSp7 tachyzoites	12/12	4/7	5/5	29/5	1/29	28/28	-
10 mg/kg RMB060	10^5^ NcSp7 tachyzoites	12/12	4/4	7/8	44/8	4/44	40/40	-
Corn oil	10^5^ NcSp7 tachyzoites	12/12	4/5	7/7	43/7	6/43	37/37	-
Corn oil	PBS	0/6	0/1	0/5	27/6	1/27	0/26	0/26

^a^ Pups were not analyzed by PCR; pups succumbing to infection postnatally are regarded as *N. caninum* positive.

## Data Availability

The data are available as Appendix A (see above).

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
