# Peer review of "In Vitro versus in Mice: Efficacy and Safety of Decoquinate and Quinoline-O-Carbamate Derivatives against Experimental Infection with Neospora caninum Tachyzoites"

_pathogens, 2023, doi:10.3390/pathogens12030447_

Round 1

Reviewer 1 Report

In vitro versus in mice: effects of decoquinate and quinoline-O- 2 carbamate derivatives against experimental infection with  Neospora caninum tachyzoites  

Type of manuscript: article 

Section: Pathogens  

General comments 
This work describes the parasitotic and parasiticidal effects of decoquinate and quinoline-O- 2 carbamate derivatives in N. caninum tachyzoites in vitro and in vivo. The study is complete and opens the door for new questions. I like it when articles like this can be found published because it is good to see that some experiments or compounds do not work, and as I said before, opening the door to make the question: Why? Can we change something? Is it worth continuing the study with these compounds?  

I‘d like to see this article published. However, to improve the understanding of the manuscript, I suggest some changes. 

Introduction 

General comments:  

  1. Need to be more precise why these are critical new drugs for the treatment and be more specific about the parasite stage in all the cases – zoites stages. For example, you mentioned Clindamycin. Why are you looking for new drugs against trophocytes if this drug is working OK – is it reported resistance or contraindications in pregnant patients?     

  1. Please, be more specific about why you used human foreskin fibroblast.  

  1. I think it will be interesting to include some lines that help the lector understand the different stages of the parasite that are important for the treatment. For example, why is more important the treatment of trophozoites vs bradyzoites?  

Line 39: Please, use a reference.  

Line 49-50: I think it connects with line 39. Please, merge these two lines and generate only one.  

Lines 38-50: It is important to say which cell types are infected by Neospora. I assume that all cells with nuclei can be infected, but why are some cell types infected more than others? Why have you used fibroblast and no other cells for the in vitro experiments?  

Line 53: Please, use a reference.  

Line 54: Treatment or treatments? I think all the drugs you mentioned are inefficient for the cysts-bradyzoites stage. Apologies, maybe the last lines are not clear to me. I understand that you can combine or not those drugs –depending on the clinical aspect of the patient.   

Lines 51-57: It is crucial to clarify that you are looking for new drugs for treating the bradyzoites stages and not for the trophozoites ones. Here, you mentioned options for the trophozoite stage, and you did not mention any disadvantage – no resistance? No refractory responses? I want to read more clearly the WHY of what you said in line 57.  

Line 64: Please, use a reference.  

Line 65: I think that, for those not in the field of pharmacology, it will be great to read what this means. Is this an advantage or disadvantage? Is this mean that the drug can penetrate all cell and tissue types? Is it related to the pharmacokinetics of the drug?  

Line 66: I wonder if the DCQ was also tested in bradyzoites? Reading this, it seems that now it is more important to look for new drugs for this stage than the trophozoite. 

Line 68: I think this is the first time you mentioned Toxoplasma. Please, write the complete name of the parasite for this time.  

Lines 71-72: Did you investigate the response in both stages of the parasite or only trophozoites?   

Lines 73-74: I assume this is connected with line 65. Could you please put it all together in the same line/s? I think it will be easier for the lector to read and understand. 

Line 78: is this after metabolization? Is the metabolization in the liver?  

Line 81-82: Trophozoites or bradyzoites?    

Line 87: Maybe in human foreskin... or infected human …  

Material and Methods 

Line 103-104: was it different for in vivo experiments?  

Line 106: as described previously by author et al., … (cite). 

Line 109: same as 106.  

Line 125-126: are the different concentrations based on previous studies?  

Line 169: Please, write forty-two to start the line.  

Section 2.9: Was it an RT-qPCR or a qPCR? It is not clear if you did a DNA extraction or RNA.  I understood that it was a qPCR using DNA and not an RT-qPCR. Please, try to write this part again. Also, you used standards to measure the parasite concentration.  

Results 

Figure E-H: The quality of these images must be improved. Also, the different references for the conditions should be more obvious. I found it a bit difficult to distinguish between both. Maybe different colours 

Reference of Fig 1: Please, be more specific to what experiments these results correspond to  

Table 1: I think these results are the same as the Figures. I think figures are more friendly to express and understand a result. Please, if these results are here to show the same/similar thing, consider moving the Table to supplementary material.  

Line 249-252: This is M&M.   

Section 3.1: You can describe the results more (e.g. Figure 1). Try not to repeat data from M&M.  

Section 3.2: Could you please add that this has been done in Human fibroblast cells?  

Figure 2: Is it possible to add to the figure the time points? I think it will be easier to follow them.  

Figure 3: The separation between the lines of the legend seems to differ from the previous ones.  

Line 300-302: this seems to be an essential result (RMB060 and the parasiticidal activity). Why is it not in the main body of the MS?  

Section 3.4: I think it could be great to read, at the very beginning of the section, why RMB054 was not tested in vivo.  

Line 305-307: M&M 

Line 348: Fig 5C? I understand that this result relates to Fig 5C? Please, if this is correct, reference it here.  

Figure 5C: This result is very strange. Does this mean the treated animals had more parasites than the non-treated ones? Am I right? Also, the error bar in the DCQ stinks. I hope to read more about this in the discussion.   

Discussion 

Line 364-358: Are you talking about tachyzoites? Could you please mention the relevant difference between ELQs and DCQs?  

Lines 370-372: I think mentioning this in the Introduction is better. 

Lines 372-373: same, because this explains why you did the chemical modifications –now I understand, but better to mention this earlier.  

Lines 377-381: Introduction. I think it is too late in MS to explain what you did. However, this should be very clear at the very beginning. 

Lines 451-471: Is it possible to discuss the different animal placentas and the impact on the pharmacokinetics/dynamics of these compounds? Could it be possible that the rumen has any implication in pharmacokinetics/dynamics?

Author Response

We thank Reviewer 1 for his critical comments on our manuscript and appreciate his effort. We here provide point-by-point responses

General comments 
This work describes the parasitotic and parasiticidal effects of decoquinate and quinoline-O- 2 carbamate derivatives in N. caninumtachyzoites in vitro and in vivo. The study is complete and opens the door for new questions. I like it when articles like this can be found published because it is good to see that some experiments or compounds do not work, and as I said before, opening the door to make the question: Why? Can we change something? Is it worth continuing the study with these compounds?  

I‘d like to see this article published. However, to improve the understanding of the manuscript, I suggest some changes. 

Introduction 

General comments:  

  1. Need to be more precise why these are critical new drugs for the treatment and be more specific about the parasite stage in all the cases – zoites stages. For example, you mentioned Clindamycin. Why are you looking for new drugs against trophocytes if this drug is working OK – is it reported resistance or contraindications in pregnant patients?    

Response: the three infectious stages of N. caninum, namely rapidly proliferating tachyzoites, tissue cyst forming bradyzoites and oocysts/sporozoites generated via a sexual process, are shortly introduced. We now also added that it is the tachyzoite stage that is responsible for acute disease symptoms. In addition, the following was added:

In lanes 61-67 we write.: For the treatment of canine neosporosis, the lincosamid clindamycin, often combined with synergistically acting sulfonamides and pyrimethamine, is the primary treatment option. However, whilst these treatments inhibit tachyzoites, they do not affect the bradyzoite stage, and effects of are most pronounced early after infection [3]. Treatments have to be taken over long periods, and for dogs with neurological signs, the prognosis is poor. In addition, clindamycin treatment in horses, cattle, sheep and goats may cause severe diarrhea that can result in death

  1. Please, be more specific about why you used human foreskin fibroblast.

Response: Human foreskin fibroblasts were used as host cells to propagate and assess drug effects against N. caninum tachyzoites. Lanes 98-101: In this study, we demonstrate the efficacy of DCQ, RMB054, RMB055 and RMB060 against N. caninum tachyzoites that are grown in human foreskin fibroblast monolayers as host cells and also study the effects of these drugs on the viability of non-infected fibroblasts. Tachyzoites represent the disease-causing stage of N. caninum.

  1. I think it will be interesting to include some lines that help the lector understand the different stages of the parasite that are important for the treatment. For example, why is more important the treatment of trophozoites vs bradyzoites?

Response: We added some information that explains the three different stages in a bit more detail (lanes 46-56): Infection can take place by ingestion of tissues infected with tachyzoites or tissue cysts, or by ingestion of food or water contaminated with sporulated oocysts. Most importantly, the temporal immunomodulation that oocurs during pregnancy allows vertical transmission of tachyzoites from the dam to its fetus, either during primo-infection in a pregnant animal (exogenous transplacental transmission), or through recrudescence of the quiescent bradyzoites that re-differentiate into tachyzoites (endogenous transplacental transmission). N. caninum tachyzoites can infect virtually any cell type, a feature they share with the closely related Toxoplasma gondii. However, in contrast to T. gondii, no cases of human infection by N. caninum have been reported so far [1]. Besides causing disease in pregnant cattle, neosporosis can lead to neonatal complications and neuromuscular diseases in dogs [2,3].

Line 39: Please, use a reference.  

Response: Done

Line 49-50: I think it connects with line 39. Please, merge these two lines and generate only one.  

Response: Done, cattle are now mentioned in line 39, dogs at the end of the paragraph (lane 56)

Lines 38-50: It is important to say which cell types are infected by Neospora. I assume that all cells with nuclei can be infected, but why are some cell types infected more than others? Why have you used fibroblast and no other cells for the in vitro experiments?  

Response: N. caninum tachyzoites infect any cell type, this information is now found on lane 52-53.

Here we use human foreskin fibroblasts for screening purposes, as this is an established methodology and has been done in many other previous publications. In addition, these fibroblasts are primary, non-transformed cell cultures. However, we could have used any other cell lines.

Line 53: Please, use a reference.  

Response: The reference is [3], mentioned right after the next sentence

Line 54: Treatment or treatments? I think all the drugs you mentioned are inefficient for the cysts-bradyzoites stage. Apologies, maybe the last lines are not clear to me. I understand that you can combine or not those drugs –depending on the clinical aspect of the patient.

Response: The reviewer is correct: it should read treatments. This is corrected (lanes 60-64).

Lines 51-57: It is crucial to clarify that you are looking for new drugs for treating the bradyzoites stages and not for the trophozoites ones. Here, you mentioned options for the trophozoite stage, and you did not mention any disadvantage – no resistance? No refractory responses? I want to read more clearly the WHY of what you said in line 57.  

Response: We here demonstrate only activity of compounds against the tachyzoite stage. The bradyzoite stage cannot be assessed in vitro. The statement on lane 57 saying that safe and efficient drugs against neosporosis in farm animals are lacking [4] clearly indicates that we are looking for safe and efficient drugs against N. caninum infection in farm animals. Therefore, we are repurposing (as mentioned in the last sentence of this paragraph) novel decoquinate derivatives and test, whether these drugs have an activity against N. caninum tachyzoites.

Line 64: Please, use a reference.  

Response: Done

Line 65: I think that, for those not in the field of pharmacology, it will be great to read what this means. Is this an advantage or disadvantage? Is this mean that the drug can penetrate all cell and tissue types? Is it related to the pharmacokinetics of the drug?  

Response: Yes, low solubility will limit absorption and bioavailability of the drug, this information was added (lane 75): DCQ has a good safety profile but is highly lipophilic and has exceedingly low aqueous solubility (0.06 μg/mL), which limits absorption and the bio-availability of the compound

Line 66: I wonder if the DCQ was also tested in bradyzoites? Reading this, it seems that now it is more important to look for new drugs for this stage than the trophozoite. 

Response: DCQ has not been tested in bradyzoites, as this is not easily done, there is no efficient culture system that allows drug testing for N. caninum bradyzoites

Line 68: I think this is the first time you mentioned Toxoplasma. Please, write the complete name of the parasite for this time.  

Response: Correct remark by the reviewer. We have now mentioned Toxoplasma in a previous paragraph (lane 54).

Lines 71-72: Did you investigate the response in both stages of the parasite or only trophozoites?  

Response: N. caninum does not make easily tissue cysts (and bradyzoites) in mice. We therefore changed the sentence as follows (lane 81-83): Thus, we investigated whether DCQ treatment would be also active in mice experimentally infected with N. caninumtachyzoites. 

Lines 73-74: I assume this is connected with line 65. Could you please put it all together in the same line/s? I think it will be easier for the lector to read and understand. 

Response: The previous paragraph was dealing with DCQ, this paragraph now deals with the DCQ derivatives. Derivatives were made due to the poor drug-like properties of DCQ. We would prefer to keep this as it is.

Line 78: is this after metabolization? Is the metabolization in the liver?

Response: in the liver

Line 81-82: Trophozoites or bradyzoites?    

Response: Tachyzoites

Line 87: Maybe in human foreskin... or infected human …  

Response: we clarified this by writing: In this study, we demonstrate the efficacy of DCQ, RMB054, RMB055 and RMB060 against N. caninum tachyzoites that are grown in human foreskin fibroblast monolayers as host cells and also study the effects of these drugs on the viability of non-infected fibroblasts. Tachyzoites represent the disease-causing stage of N. caninum.

Material and Methods 

Line 103-104: was it different for in vivo experiments?  

Response: As indicated in section 2.8, the drugs to be applied in in vivo studies were suspended in corn oil. We also indicate under 2.1: … for in vivo experiments, compounds were resuspended in corn oil at concentrations indicated below.

Line 106: as described previously by author et al., … (cite). 

Response: now lane 121: …..were cultured as described by Ramseier et al [12].

Line 109: same as 106.  

Response: now lane 124….Winzer et al

Line 125-126: are the different concentrations based on previous studies?  

Response: yes, as indicated

Line 169: Please, write forty-two to start the line.  

Response: The sentence was changed (now lane 181)

Section 2.9: Was it an RT-qPCR or a qPCR? It is not clear if you did a DNA extraction or RNA.  I understood that it was a qPCR using DNA and not an RT-qPCR. Please, try to write this part again. Also, you used standards to measure the parasite concentration.  

Response: We state at the beginning of the paragraph that we purified DNA, thus we did qPCR. The reviewer has correctly noted a mistake in the description of the method, which has now been corrected in section 2.9.

Results 

Figure 1 E-H: The quality of these images must be improved. Also, the different references for the conditions should be more obvious. I found it a bit difficult to distinguish between both. Maybe different colours? 

Response: The resolution of the graphs has been improved to increase visibility (Fig. 1 E-H)

Reference of Fig 1: Please, be more specific to what experiments these results correspond to.   

Table 1: I think these results are the same as the Figures. I think figures are more friendly to express and understand a result. Please, if these results are here to show the same/similar thing, consider moving the Table to supplementary material.  

Response: Actually Fig. 1 E-H and Table 1 do not show the same results, but corresponding results. Figure 1 shows dose-responses of tachyzoite proliferation in response to different dosages of the drugs, Table 1 shows the concentrations at which 50% proliferation inhibition occurred. Hence, we added the following sentence into the legend of Figure 1: “The corresponding IC50 values are listed in Table 1”.

Line 249-252: This is M&M.   

Response: Correct, we deleted this part

Section 3.1: You can describe the results more (e.g. Figure 1). Try not to repeat data from M&M.  

Response: the text was slightly modified by deleting materials and methods related aspects.

Section 3.2: Could you please add that this has been done in Human fibroblast cells?  

Response: Done: 3.2. Structural alterations induced by in vitro drug treatments of N. caninum infected HFF

Figure 2: Is it possible to add to the figure the time points? I think it will be easier to follow them.  

Response: Done

Figure 3: The separation between the lines of the legend seems to differ from the previous ones.  

Response: Correct, but we don’t know how to fix this

Line 300-302: this seems to be an essential result (RMB060 and the parasiticidal activity). Why is it not in the main body of the MS?

Response: Correct that this is an essential result. However, for the long-term treatments we show photomicrographs in all cases, and we don’t think it makes a lot of sense to place these into the manuscript asa figure. We will of course do this in case the reviewer insists.

Section 3.4: I think it could be great to read, at the very beginning of the section, why RMB054 was not tested in vivo.  

Response: Thank you for the suggestions. We started the paragraph as follows: “Since long term treatments showed that RMB054 did not act parasiticidal, this derivative was excluded from further assessments in vivo. Thus, the efficacy of RMB060 was assessed comparatively with DCQ in pregnant and non-pregnant N. caninum infected BALB/c mice.

Line 305-307: M&M 

Response: Correct, we deleted the text referring to the methodology

Line 348: Fig 5C? I understand that this result relates to Fig 5C? Please, if this is correct, reference it here.  

Response: this is a mistake, it should read Figure 4C, not 5C. Sorry for the error.

Figure 5C: This result is very strange. Does this mean the treated animals had more parasites than the non-treated ones? Am I right? Also, the error bar in the DCQ stinks. I hope to read more about this in the discussion.   

Response: we can only report the results we actually get, no matter whether they look strange or not. Fig. 4C refers to parasite loads in dams. The error bar that “stinks” is mostly due to one mouse that had to be euthanized earlier, as indicated in the text and the reason this animal exhibited severe clinical signs was most likely due to increased parasite load. However, while it looks like that DCQ treated animals had higher cerebral parasite levels than the controls, the difference was not statistically significant. The message is that there is no real effect of the drug.

Discussion 

Line 364-358: Are you talking about tachyzoites? Could you please mention the relevant difference between ELQs and DCQs?  

Response: we added the information on what invasive stages were investigated. ELQs and decoquinate are closely related in that they both have a quinolone core structure, but ELQs contain additional quinones attached. Going into the chemistry is not within the scope of this paper.

Lines 370-372: I think mentioning this in the Introduction is better. 

Response: we already explained in the introduction that DCQ inhibits the cytochrome bc1 complex and in the discussion we actually try to explain what is the postulated mechanism and why it would harm the parasite. We would like to keep this as part of the discussion.

Lines 372-373: same, because this explains why you did the chemical modifications –now I understand, but better to mention this earlier.  

Response: The rational for making these modifications are given in the Introduction lanes 84-89:

Due to the poor drug like properties of DCQ, derivatives with enhanced water solubility were generated by adding highly polar groups to the initial compound [9,10]. To treat systemic infections more successfully, in our case, DCQ was derivatized to the quinoline-O-carbamate derivatives RMB054, RMB055 and RMB060 (Fig. 1B-D). The addition of polar groups was thought to increase absorption, with the appendages metabolized in the liver following uptake and resulting in increased DCQ levels after that.

Lines 377-381: Introduction. I think it is too late in MS to explain what you did. However, this should be very clear at the very beginning. 

Response: This part was deleted

Lines 451-471: Is it possible to discuss the different animal placentas and the impact on the pharmacokinetics/dynamics of these compounds? Could it be possible that the rumen has any implication in pharmacokinetics/dynamics?

Response: we had thought about this point as well, as the mouse and ruminant placentas exhibit clear differences. However, we do also not see any favourable effect in non-pregnant mice, thus we don’t think that the differences are due to placental differences. It is, however, possible that the rumen has implications in pharmacokinetics/dynamics, but this would be speculation and is not backed up by any data in the literature, thus we don’t want to embark on this discussion.

Reviewer 2 Report

A well conducted study of aspects of the safety and efficacy of potential treatments for N caninum infection in vitro and in vivo.

Only minor comments:

Should the title talk of effects or "efficacy and safety"?

Line 56, is "alternatively" justified, why not just delete?

Line 87 - is it "and"? or "in" human .....?Strain Nc-Sp7 might need some introduction - it just gets mentioned suddenly in Line 123

Figure 4 A - the lines are not clearly distuinguishable in my pdf? can that be improved - particularly for the two C lines.

Line 474 - is that as a summary statement warranted for all in vitro characteristics?

Line 511 - "addressed" - an odd turn of phrase in English - " are due"?

Author Response

We thank Reviewer 2 for his positive assessment and his comments.

Should the title talk of effects or "efficacy and safety"?

Response: good point, we replaced “effects” by “efficacy and safety”

Line 56, is "alternatively" justified, why not just delete?

Response: good point, “alternatively” is now deleted.

Line 87 - is it "and"? or "in" human .....?Strain Nc-Sp7 might need some introduction - it just gets mentioned suddenly in Line 123

Response: the entire sentence has been changed (lanes 98-101): In this study, we demonstrate the in vitro efficacy of DCQ, RMB054, RMB055 and RMB060 against N. caninum tachyzoites that are grown in human foreskin fibroblast monolayers as host cells, and also study the effects of these drugs on the viability of non-infected fibroblasts.

Figure 4 A - the lines are not clearly distuinguishable in my pdf? can that be improved - particularly for the two C lines.

Response: we increased the size of the figure to make it more clear

Line 474 - is that as a summary statement warranted for all in vitro characteristics?

Response: The sentence was changed as follows: Despite being highly efficacious in vitro (IC50 values in the low nanomolar range and low host cell toxicity) and parasiticidal activity noted for RMB060, both DCQ and RMB060 failed to limit parasite dissemination and fetal infection in the pregnant neosporosis mouse model.

Line 511 - "addressed" - an odd turn of phrase in English - " are due"?

Response: the sentence was changed, “addressed” deleted

Round 2

Reviewer 1 Report

Thank you very much for including my suggestions. In my opinion, the Ms is more clear and more ready to be published. Just be careful that there are some comments (probably using Microsoft word). 

 Congratulations!